# A structural model of flagellar filament switching across multiple bacterial species

Fengbin Wang[1], Andrew M. Burrage [2], Sandra Postel[3], Reece E. Clark[2], Albina Orlova[1], Eric J. Sundberg [3,4], Daniel B. Kearns[2] & Edward H. Egelman [1]

The bacterial flagellar filament has long been studied to understand how a polymer composed of a single protein can switch between different supercoiled states with high cooperativity. Here we present near-atomic resolution cryo-EM structures for flagellar filaments from both Gram-positive *Bacillus subtilis* and Gram-negative *Pseudomonas aeruginosa*. Seven mutant flagellar filaments in *B. subtilis* and two in *P. aeruginosa* capture two different states of the filament. These reliable atomic models of both states reveal conserved molecular interactions in the interior of the filament among *B. subtilis*, *P. aeruginosa* and *Salmonella enterica*. Using the detailed information about the molecular interactions in two filament states, we successfully predict point mutations that shift the equilibrium between those two states. Further, we observe the dimerization of *P. aeruginosa* outer domains without any perturbation of the conserved interior of the filament. Our results give new insights into how the flagellin sequence has been "tuned" over evolution.

[1] Department of Biochemistry and Molecular Genetics, University of Virginia School of Medicine, Charlottesville, VA 22908, USA. [2] Department of Biology, Indiana University, Bloomington, IN 47305, USA. [3] Institute of Human Virology and University of Maryland School of Medicine, Baltimore, MD 21201, USA. [4] Departments of Medicine and of Microbiology & Immunology, University of Maryland School of Medicine, Baltimore, 21201 MD, USA. Fengbin Wang and Andrew M. Burrage contributed equally to this work. Correspondence and requests for materials should be addressed to E.H.E. (email: egelman@virginia.edu)

Most motile bacteria have complex structures known as flagella, with extracellular filaments that grow up to 15 μm long and spin at hundreds of revolutions per second. There are three parts to the flagellar structure: the trans-membrane basal body that functions as the motor, the connecting rod and hook, and the flagellar filament that acts like a helical propeller[1]. The bacterial flagellar filament has been intensively studied for many years[2–4]. It has served as an enlightening system for understanding how a protein polymer composed of a single protein, flagellin (except for the cap protein at the end that acts as an assembling chaperone[5, 6]) switches among different states to supercoil. This supercoiling allows the rotating filament to behave as an Archimedean screw and produce thrust. The filament can adopt different conformational states due to mechanical forces, such as when the motor switches the sense of rotation[7], allowing the bacteria to swim forward, backward, in a screw-like fashion and to tumble[8]. With the motor linked to sensory receptors[9], the bacteria are capable of moving towards nutrients and away from dangerous environments, resulting in a significant survival advantage[10]. On the other hand, mutations within the flagellin protein that fail to form supercoiled filaments generate no thrust when such straight filaments are rotated, leading to non-motile bacteria[11–13].

Our current understanding of supercoiling in bacterial flagellar filaments, referred to as polymorphic switching, is based upon the notion that protofilaments within the flagellar filament can exist in two discrete states, that differ slightly in length[14–22]. The filaments curve due to shorter protofilaments forming the inside of supercoils, with longer protofilaments on the outside, generating periodic waveforms. Given 11 protofilaments in the intensively studied *Salmonella enterica* flagellar filaments, 12 different supercoiled states have been proposed, ranging from all 11 protofilaments in the "short" state to all 11 protofilaments in the "long" state. The "short" state results from a protofilament having a right-handed inclination (the R-state), while the "long" state results from a protofilament having a left-handed inclination (the L-state)[23]. The structure of wild-type flagellar filaments with non-straight waveforms cannot be analyzed at high resolution easily, because the filaments do not have a simple helical symmetry in which every subunit is in an equivalent environment. To reconstruct filaments at high resolution, all protofilaments must be "locked" into the same state, either L- or R-type, producing straightened filaments that lead to non-motile bacteria. Based upon extensive work from the Namba laboratory[23–26] using X-ray fiber diffraction, X-ray crystallography, and cryo-EM, atomic models have been proposed for straight *Salmonella enterica* filaments with all protofilaments in either the R-state[25] or the L-state[24].

While these two atomic models represent a significant advance in understanding polymorphic switching of bacterial flagellar filaments, they do not provide sufficient mechanistic understanding of how switching occurs. Indeed, these models raise numerous questions that will need to be addressed so that we can develop drugs to inhibit the flagellar functions of pathogenic bacteria and engineer novel nano-machines for controlled movement of molecular cargoes. First, each model is based upon only a single amino acid variant, which came from the selection of flagellin mutants that cause loss of motility. Whether other motility mutants lead to the same or similar atomic models, or if a multiplicity of different states might arise from different mutants is unclear. Second, it has emerged that there is a divergence of quaternary structure in bacterial flagellar filaments, with those from *Campylobacter jejuni* having seven, rather than 11 protofilaments[27]. As *C. jejuni* and *Salmonella* are both Gram-negative bacteria, the degree of divergence among not only Gram-negative organisms but also among the Gram-positive populations has not

been resolved, as is the question of whether Gram-positive bacteria share a conserved flagellar filament structure with Gram-negative bacteria. Third, enormous advances have been made in resolution using cryo-EM over the past 4 years, largely driven by the availability of direct electron detectors[28]. These detectors were not available when the *Salmonella* models were proposed[24, 25]. Here, we attempt to answer these questions and others using a direct electron detector by generating and studying locked, straight flagellar filaments from the Gram-positive bacterium *B. subtilis* and the Gram-negative *P. aeruginosa* with high resolution cryo-EM.

## Results

**Cryo-EM structures of *B. subtilis* flagellar filaments.** Flagellin protein in *Bacillus subtilis* is expressed from the *hag* gene[29], which is homologous to *fliC* in *Salmonella enterica*. To obtain straight flagellar filaments suitable for cryo-EM analysis, we used low-fidelity PCR to randomly mutagenize an allele of the *hag* gene, $hag^{T209C}$, and screened for non-motile mutants in *B. subtilis*. The non-motile mutants could be fluorescently labeled with a maleimide dye due to the $Hag^{T209C}$ allele and were assessed by fluorescence microscopy to identify straight structures (Blair et al. 2008). Of the mutants screened, 30 were non-motile and, of those, six were determined to have straight flagellar filaments by fluorescent microscopy. We also included a previously identified straight filament mutant, $Hag^{A233V}$, in the study[30]. Flagella were isolated from the mutants and cryo-EM images were collected using a Falcon II direct electron detector (Fig. 1a). Filaments were boxed and cut into overlapping segments, filament segments were sorted by the considerable structural polymorphism (both rise and rotation), and reconstructed to near-atomic resolution (the helical parameters and statistics of all seven reconstructions are listed in Table 1). Notably, all seven mutants (three L-type and four R-type mutants) have right-handed 1-start, left-handed 5-start and right-handed 6-start helices (Figs. 1b, c, Supplementary Fig. 1a, b). While the three L-type mutants (E115G, S285P, and S17P) have a left-handed 11-start helix, the four R-type mutants (N226Y, A233V, H84R and A39VN133H) have a right-handed 11-start helix, and the designations L and R correspond to the hand of these 11-start helices. The L-type 11-start helices are tilted left by ~ 1.7° and the R-type 11-start helices are tilted right by ~ 3.9° (Figs. 1b–e).

We obtained near-atomic resolution reconstructions for both L- and R-type straight filament mutants. The first atomic filament model (R-type H84R) was built *de novo* using an established Rosetta protocol[31], and the other filament models were built by RosettaCM[32] using the H84R model as the starting template. The best resolution reached for L- and R-type mutants reconstructions are 4.5 and 3.8 Å, respectively (Figs. 1d, e, Table 1) based on Fourier shell correlation (FSC) of the cryo-EM density map with the resulting filament model (Supplementary Fig. 2). We confirmed that this model:map FSC yields a similar estimate of resolution compared with the more traditional "gold standard" FSC between two independent map reconstructions (map:map). We calculated both FSC plots in the A39VN133H dataset and found that the model:map FSC (4.3 Å) is similar but more conservative compared with the map:map FSC (4.1 Å, Supplementary Fig. 2f). A top view of these reconstructions clearly shows that both L- and R-type mutants have 11 protofilaments, forming annular tubes with inner diameters of ~ 25 Å and outer diameters of ~ 125 Å (Figs. 1d, e), and these dimensions are similar to *Salmonella* if one excludes the D2/D3 domains[24, 25]. The Hag subunit is composed of two domains labeled D0 and D1, which are arranged on the inside and outside, respectively, of the flagellar filaments (Figs. 1d, e). For all seven mutants, the filament

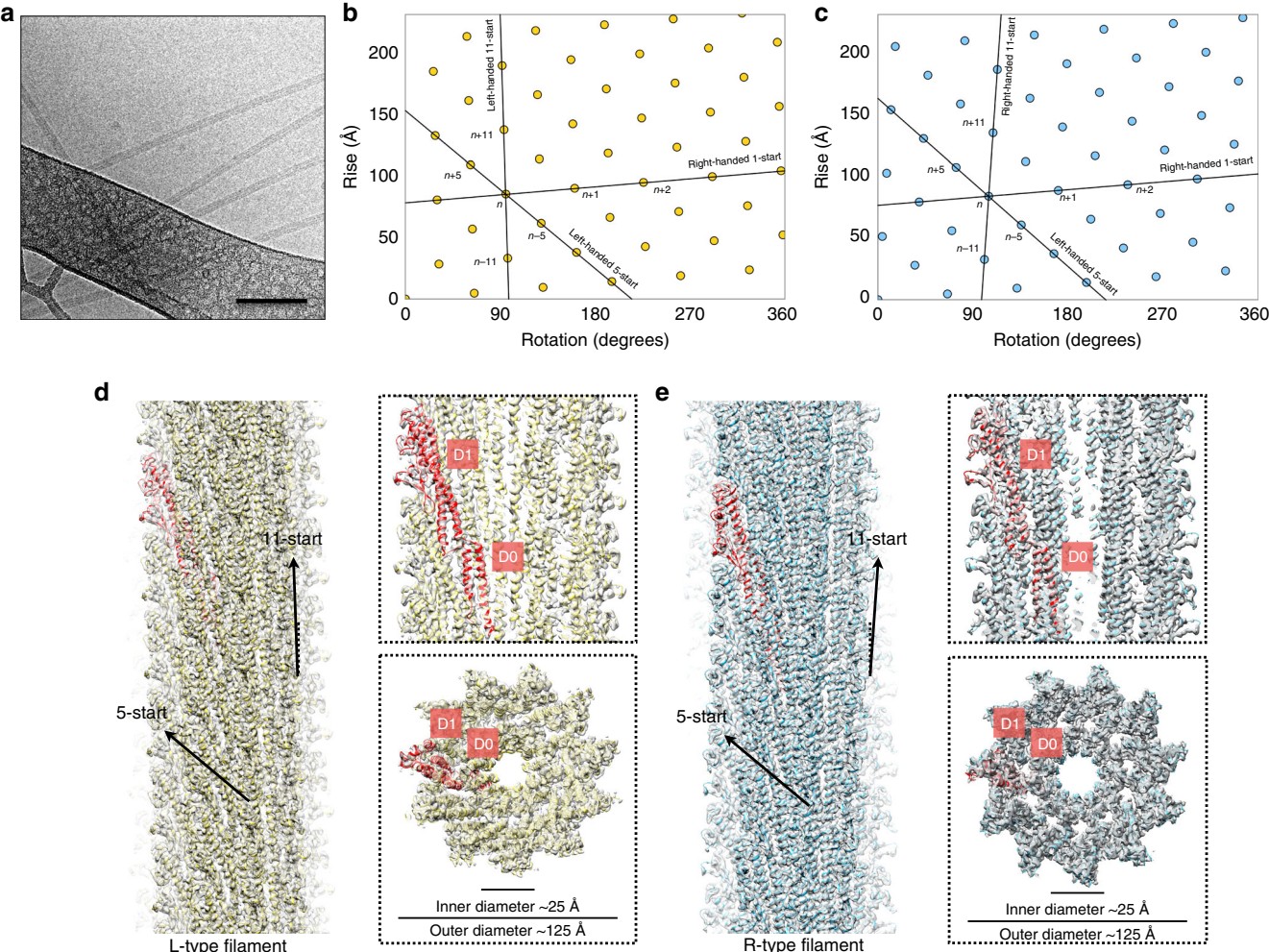

**Fig. 1** Cryo-EM reconstruction and flagellar filament model of *B. subtilis*. **a** Cryo-electron micrograph of *B. subtilis* flagellar filaments. The *scale bar* represents 100 nm. **b** and **c** The helical net of the *B. subtilis* flagellar filament using the convention that the surface is unrolled and we are looking from the outside. The L-type filaments are shown in **b** and the R-type filaments are shown in **c**. **d** and **e** The surfaces of the side view, the central slice through the lumen, and the top view of the cryo-EM reconstructions (L-type straight filament S285P shown in **d**, R-type straight filament N226Y shown in **e**)

core (D0) has a slightly better resolution than the outer region (D1), presumably due to packing stability (Supplementary Fig. 2). These observations are similar to previous reports of other helical assemblies where the outer residues are less ordered than the inner ones[33].

The subunits of L-type and R-type Hag share highly similar secondary architecture (Supplementary Fig. 3a, b). The D0 domain is composed of two α-helices (ND0 and CD0) that form a short coiled-coil. The D1 domain is composed of a β-hairpin and three α-helices (ND1a, ND1b and CD1), which form a longer coiled-coil than in D0. The D0 and D1 domains are connected by two loop regions: residues S31-D47 (NL) connect ND0 and ND1a and residues R264-D267 (CL) connect CD0 and CD1 (Supplementary Fig. 3a, b). The entire backbone, except for the first four residues at the N-terminus, can be traced unambiguously and built in the reconstruction for both L- and R-type subunits. Side chain densities can be seen for most of the residues (Supplementary Fig. 3a, b), which allows us to accurately determine the register of the amino acid sequence in the cryo-EM density. These are the highest resolution reconstructions of bacterial flagellar filaments reported to date.

**Cryo-EM structures of *P. aeruginosa* flagellar filaments.** Next, we analyzed flagellar filaments from the Gram-negative

bacterium *Pseudomonas aeruginosa* to compare with a Gram-negative species. Since the D0 and D1 domains of *P. aeruginosa* and *S. enterica* share a high degree of sequence identity (55%), we tested the conserved L- and R-type mutations characterized in *S. enterica* (G426A and A449V, respectively)[24, 25], which correspond to G420A and A443V in *P. aeruginosa*, respectively. We found that these mutations also generate straightened filaments in this organism. Filaments from these two FliC mutants of *P. aeruginosa* were sheared off the bacteria, concentrated, plunge-frozen and cryo-EM imaged (Fig. 2a). Each mutation results in L- and R-type straight filaments as expected, which suggests that the D0/D1 architecture is indeed conserved between *P. aeruginosa* and *S. enterica*.

For both the L- and R-type *P. aeruginosa* filaments, we identified three pseudo layer-lines in the power spectra from the filaments, besides the 1-, 5-, 6- and 11-start layer-lines that are similar to *B. subtilis*, which correspond to non-helical perturbations (Supplementary Fig. 1c, d). Similar perturbations have been observed before in a particular *Salmonella* strain[34] as well as in other bacteria[35–39], and correspond to a larger asymmetric unit containing two flagellin subunits. The non-helical nature of the pairing is due to the fact that a seam is introduced into the structure, which is a discontinuity in the helical surface lattice (Fig. 2c). To further investigate whether this non-helical

**Table 1 Refinement statistics for the flagellar filament models**

| Mutation site (s) | B. subtilis | | | | | | | P. aeruginosa | |
|---|---|---|---|---|---|---|---|---|---|
| | S285P | E115G | S17P | N226Y | A39VN133H | H84R | A233V | G420A | A443V |
| Filament hand (11-start) | Left | Left | Left | Right | Right | Right | Right | Left | Right |
| *Helical symmetry* | | | | | | | | | |
| Rise (Å) | 4.72 | 4.72 | 4.68 | 4.64 | 4.65 | 4.64 | 4.64 | 4.73 | 4.61 |
| Rotation (°) | 65.30 | 65.30 | 65.29 | 65.83 | 65.81 | 65.81 | 65.81 | 65.27 | 65.75 |
| CTF selected images | 569 | 285 | 539 | 604 | 826 | 490 | 324 | 104 | 637 |
| Total segments | 138,327 | 41,587 | 67,195 | 97,487 | 315,847 | 165,589 | 74,183 | 17,450 | 209,965 |
| Sorted segments | 55,403 | 22,682 | 13,899 | 72,005 | 134,766 | 58,771 | 33,992 | 17,450[b] | 102,119 |
| Resolution[a] (Å) | 4.5 | 5.7 | 6.7 | 3.8 | 4.3 | 4.4 | 5.5 | 4.3 | 4.2 |
| Clash score, all atoms | 6.3 | 3.7 | 3.1 | 4.3 | 3.3 | 1.9 | 4.0 | 9.1 | 3.6 |
| *Protein geometry* | | | | | | | | | |
| Ramachandran favored (%) | 88.9 | 90.5 | 87.3 | 91.0 | 90.7 | 91.7 | 90.6 | 93.5 | 90.2 |
| Ramachandran outliers (%) | 0 | 0 | 0 | 0 | 0 | 0 | 0 | 0 | 0 |
| Rotamer outliers (%) | 0 | 0 | 0 | 0 | 0 | 0 | 0.4 | 0 | 0 |
| Cβ deviations > 0.25 Å | 0 | 0 | 0 | 0 | 0 | 0 | 0 | 0 | 0 |
| *RMS deviations* | | | | | | | | | |
| Bond (Å) | 0.006 | 0.007 | 0.007 | 0.008 | 0.005 | 0.007 | 0.006 | 0.010 | 0.008 |
| Angels (°) | 0.98 | 1.16 | 1.03 | 1.11 | 0.88 | 0.96 | 0.96 | 1.36 | 1.11 |
| Molprobity score | 1.93 | 1.70 | 1.72 | 1.73 | 1.65 | 1.44 | 1.72 | 1.91 | 1.69 |
| Molprobity percentile | 99 | 99 | 99 | 99 | 99 | 99 | 99 | 99 | 99 |
| PDB ID | 5WJY | 5WJZ | 5WJX | 5WJT | 5WJU | 5WJV | 5WJW | 5WK6 | 5WK5 |
| EMDB ID | 8852 | 8853 | 8851 | 8847 | 8848 | 8849 | 8850 | 8856 | 8855 |

[a]Resolution of all filaments was estimated by model-map FSC (0.38 cutoff). This approach was validated by using a map:map FSC from non-overlapping data sets for the A39VN133H mutation, starting with volumes filtered to 10 Å resolution as starting references. This yielded an FSC at 0.138 of 4.1 Å
[b]Reconstruction of sorted *P. aeruginosa* G420A segments gave a worse resolution than unsorted segments, presumably due to limited number of total segments

perturbation occurs in wild-type filaments, and is not introduced by the mutations, a power spectrum of wild-type filaments was generated from 50 images of negatively stained filaments (Supplementary Fig. 1e, f) and clearly shows a layer line corresponding to this non-helical perturbation.

Ignoring the non-helical perturbation, it is possible to reconstruct these mutants by assuming there is a strict helical symmetry, with every subunit being equivalent (Fig. 2b). We obtained near-atomic resolution reconstructions for the D0 and D1 domains for both mutants: at 4.2 Å resolution for L-type filament G420A and at 4.3 Å resolution for R-type filament A443V based on FSC of the cryo-EM density map with the resulting filament model (Supplementary Fig. 2). The filament model was built by RosettaCM[32] using the *Bacillus* mutants described above as the starting model. That we could reach such a resolution for D0 and D1 confirms that the perturbation is entirely limited to the outer D2/D3 domains, as previously suggested from lower resolution studies of other bacteria[34–40].

To visualize the outer domain structure of *P. aeruginosa* filaments, the non-helical perturbation needs to be considered. Previous work on a perturbed *Salmonella* flagellar filament[34] suggested that subunits pair across the 5-start helices, and therefore a seam forms between two 11-start protofilaments (Fig. 2c). The presence of this seam, similar to the seam observed in microtubule filaments[41], breaks the continuity along the 5-start helices. While structures such as this with a seam are referred to as "non-helical", it can be seen from Fig. 2c that the structure can actually be viewed as an ideal helix, but one with an asymmetric unit containing 22 flagellin subunits. This very large asymmetric unit would be related to adjacent ones by an axial rise of 22*4.61 Å = 101.5 Å and a rotation of 6.41°, generating a 1-start helix with a pitch of 5700 Å having 56.16 units per turn. These symmetry parameters were applied (Fig. 2d) to an IHRSR reconstruction. After one cycle, inter-subunit pairing can be seen clearly in the reconstructed volume across the 5-start helices (Figs. 2c, d),

where the D3 domain of the subunit $S_N$ (green dots in Fig. 2c) interacts with the D2 domain of subunit $S_{N+11}$ (red dots in Fig. 2c). The pseudo layer-lines reflecting this pairing can be seen from the power spectrum of the projection of this volume (Supplementary Fig. 4), matching the layer lines seen in the actual images. Unfortunately, the reconstruction fell apart very quickly with additional cycles so a higher resolution reconstruction could not be obtained. This was not surprising and is due to the very large asymmetrical unit combined with the poor signal-to-noise ratio in the micrographs.

**Overall comparison of the L- and R-type structures.** Since these are the first near-atomic resolution structures of L- and R-type filaments from both Gram-positive and -negative strains, we set out to compare detailed differences in the molecular architecture between L- and R-type filaments. First, we compared a single subunit from each filament type by superimposing their $C_\alpha$ backbones (Fig. 3a), which shows that the L- and R-type subunits of *B. subtilis* share the same secondary structure. When two subunits are aligned by their D0 domains, which comprise the inner part of the filament, the D1 domain of the L-type subunit was twisted clockwise from the D1 domain of the R-type subunit (Fig. 3a). The rotation and shift for the $C_\alpha$ atoms in the D1 domain range from 2 to 10° and 1.5–11 Å, respectively. The D1 domain $C_\alpha$ atoms twist more when they are further from the D0 domain, which places them at the outer region of the filament. A similar twisting motion was observed when L- and R-type subunits were aligned by the D1 domain. The small RMSD of individual domain D0 or D1 backbones (all <1 Å), suggests a highly conserved fold between Gram-positive and -negative bacteria, and between the L- and R-states (Fig. 3b, bottom left). However, if the RMSD is calculated from the connected D0-D1 domains, we can see a clear clustering by handedness: domains of the same hand from two species share a very conserved backbone

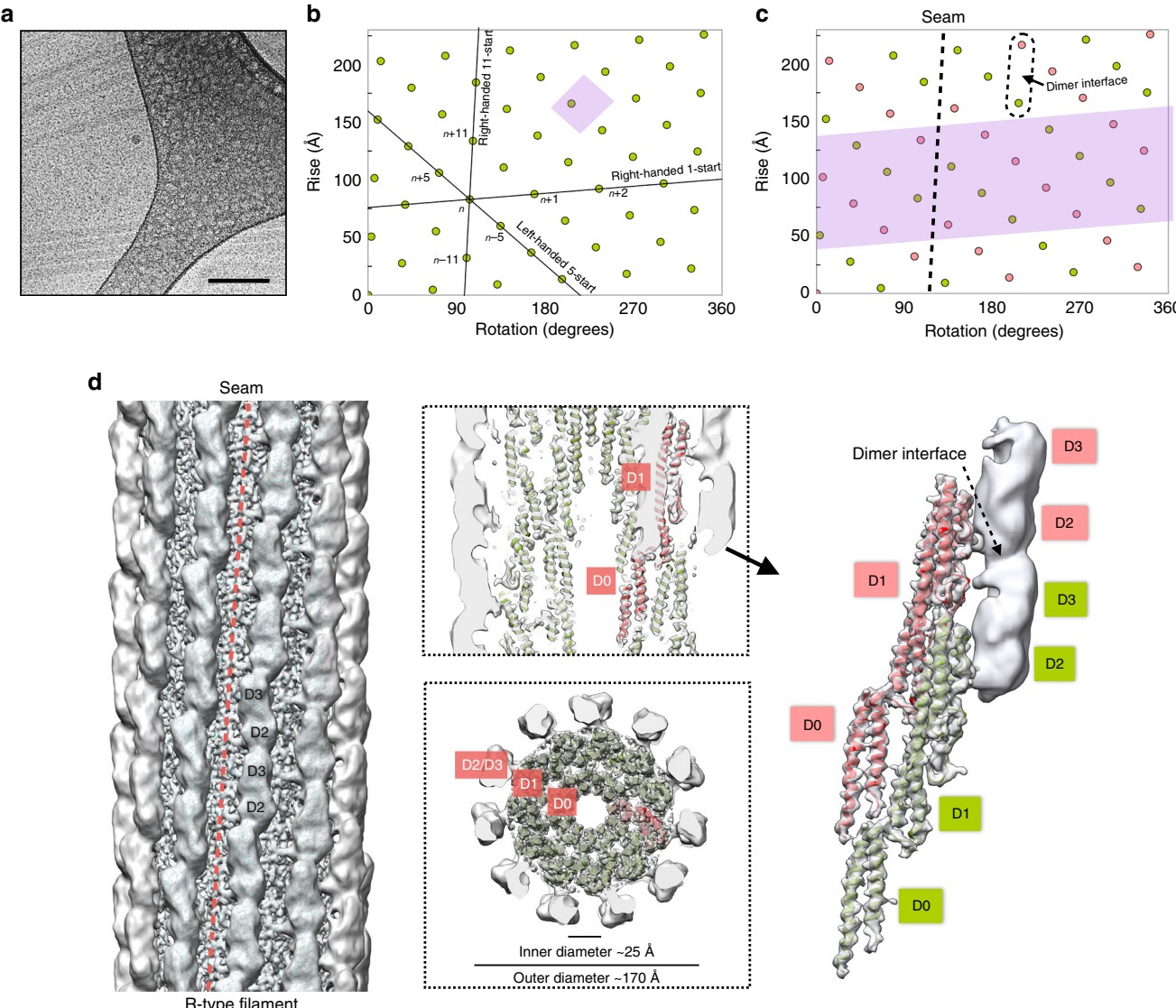

**Fig. 2** Cryo-EM reconstruction and flagellar filament model of *P. aeruginosa*. **a** Cryo-electron micrograph of *P. aeruginosa* flagellar filament. The *scale bar* represents 100 nm. **b** The helical net of R-type flagellar filament core (D0 and D1 domain) of *P. aeruginosa*. **c** The helical net of D2/D3 domain in the R-type flagellar filament of *P. aeruginosa*. **d** The side view, the central slice through the lumen, the top view and the segmented dimer view of the cryo-EM reconstructions of the R-type flagellar filament A443V of *P. aeruginosa*

trace (RMSD 0.6 ~ 0.9 Å), whereas the RMSD of different hands ranges from 1.5 to 3.3 Å (Fig. 3b, upper right). Altogether this structural investigation reveals strong conservation in the individual D0 and D1 fold between L-type and R-type filaments, and even between different bacterial species. Furthermore, it shows that the major difference between L- and R-type subunits resides in the rotation angle mediated by the two connecting loops (NL and CL).

Next, we compared the subunit-subunit interactions of the L- and R-type filaments. A subunit $S_0$ in the flagellar filament interacts with eight other subunits: $S_{+5}$, $S_{+6}$, $S_{+11}$, $S_{+16}$, $S_{-5}$, $S_{-6}$, $S_{-11}$, $S_{-16}$. Since the interactions between $S_0$ and $S_{+n}$ are the same as the interactions between $S_{-n}$ and $S_0$, only five subunits ($S_0$, $S_{+5}$, $S_{+6}$, $S_{+11}$, $S_{+16}$) were used to analyze all the unique interactions in the flagellar filaments as shown in Fig. 3c. PISA interface analysis[42] shows that for both L- and R-type structures, $S_0$ is making major contacts with $S_{+5}$ and $S_{+11}$ with an interfacial area of ~ 1900 Å², minor contacts with $S_{+16}$ with an interfacial area of ~ 270 Å², and intermediate contacts with $S_{+6}$ with an interfacial area of ~ 600 Å² (Supplementary Table 1). In the 5-start interface,

we detected contacts of $S_0$-ND1a/b to $S_{+5}$-CD1 and $S_0$-ND1b to $S_{+5}$-β-hairpin. In the $S_0$ and $S_{+11}$ interface, we detected that the $S_0$-CD1 makes contacts with $S_{+11}$-ND0, ND1 and β-hairpin, and $S_0$-ND1a interacts with $S_{+11}$-ND1 and β-hairpin. There are also contacts between $S_0$-CD0 and $S_{+11}$-ND0 and CD0. Even in the relatively smaller interface of $S_0$ and $S_{+16}$, we were able to detect interactions between D1 domains: $S_0$-ND1b and $S_{+16}$-ND1a. The interface of $S_0$ and $S_{+6}$, on the other hand, only involves the interactions of D0 domains: $S_0$-CD0 and $S_{+6}$-ND0 and CD0 (Fig. 3c, d). These observations differ from previous conclusions derived from *Salmonella* filament structures, in which the 5-start interface has been considered as the only inter-subunit D1 domain interaction and the only important interface for the L-/R- switch mechanism[24]. By analyzing homologous flagellar filaments reconstructions with much higher resolution from two other bacterial species, we suggest that the D1 domain interactions along the 5-start helices in *Salmonella* exist but are not the only interactions.

Finally, we compared the filament packing of all the L- and R-type structures by calculating the RMSD of the five subunits

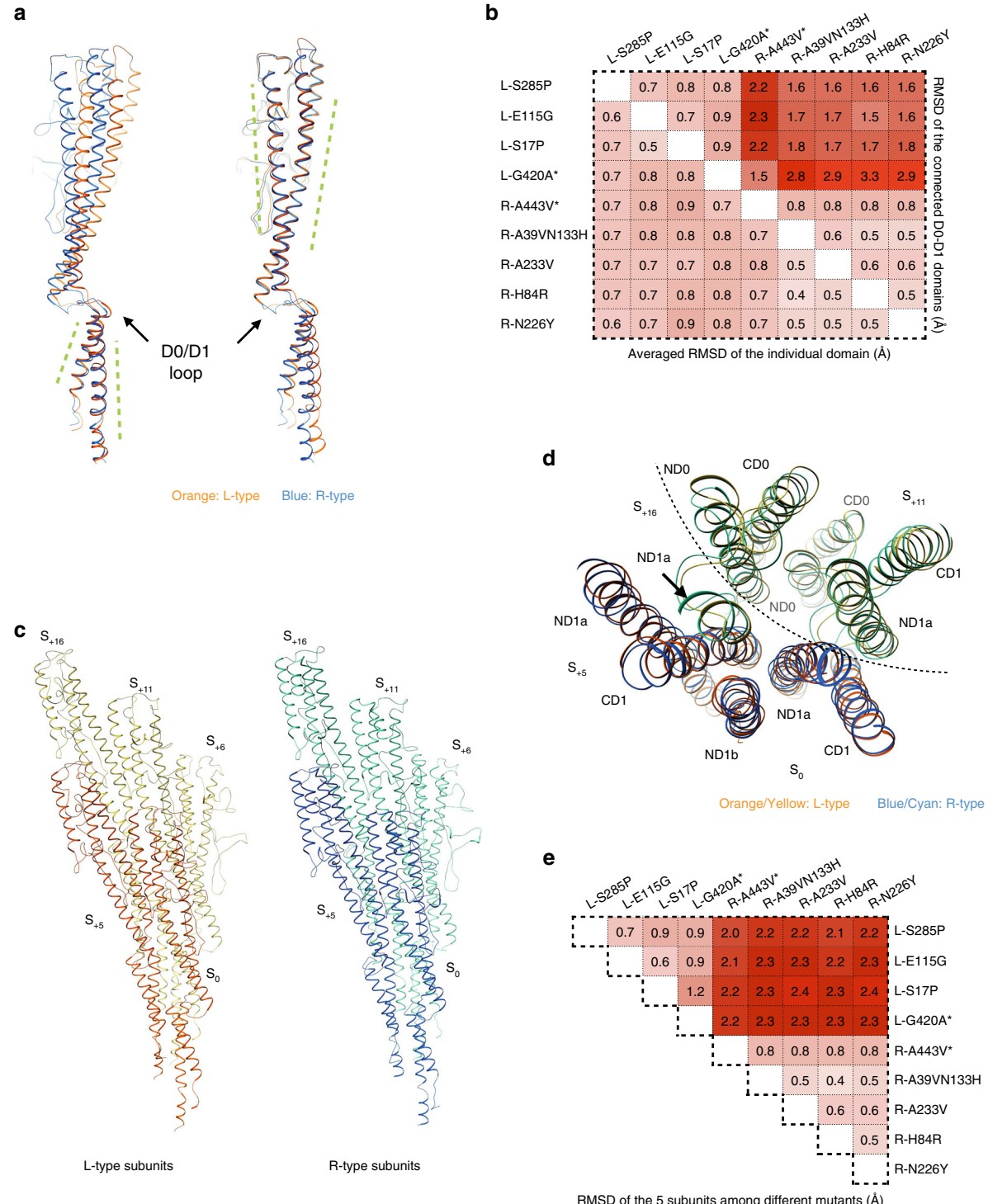

**Fig. 3** Comparison of the $C_\alpha$ backbones of L- and R- type structures. **a** Superposition of a single L-type (*orange*, S285P) and R-type (*blue*, N226Y) subunit of *B. subtilis*, by D0 domain (*left*) and D1 domain (*right*). **b** RMSD calculations of individual domains (*bottom left*) and RMSD calculations of the combined D0 plus D1 domains (*upper right*). Two *P. aeruginosa* mutants are labeled with an *asterisk*. The intensity of *red* corresponds to the RMSD level. **c** Five subunits of L-type (*yellow* and *orange*, S285P) and R-type (*blue* and *cyan*, N226Y) subunits, containing all unique contacts within the filaments. **d** Comparison of the relative arrangement of these five subunits from the horizontal plane (*yellow* and *orange*: L-type S285, *blue* and *cyan*: R-type N226Y). **e** RMSD calculations of the complex of these five subunits among all the nine filament mutants

containing all of the unique interactions (Fig. 3e). Similar to single subunit RMSD calculation (Fig. 3b), we observed striking clustering/conservation among filaments of the same hand regardless of the species. In fact, the average five-subunit RMSD of R-type filaments between two species is

only 0.8 Å, significantly lower than the average five-subunit RMSD (greater than 2.0 Å) between any L- and R-type filaments (Fig. 3e). These calculations strongly suggest that filaments of the same hand are not only conserved in their single subunit architecture, but also highly conserved in terms of

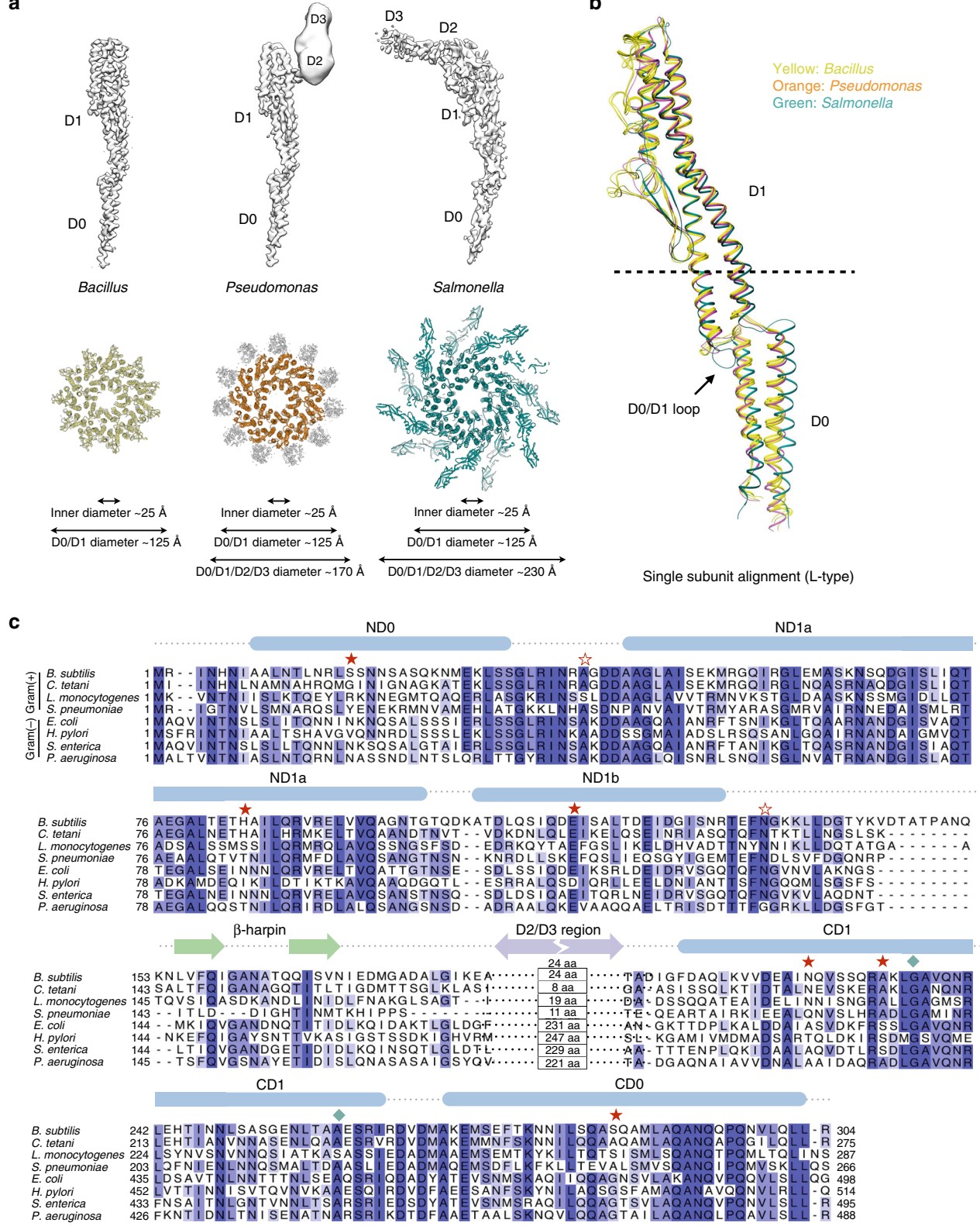

**Fig. 4** Comparison with flagellar filaments of other bacterial species. **a** A comparison of L-type flagellar structures in *B. subtilis* (S285P), *P. aeruginosa* (G420A) and *S. enterica* (PDB: 3A5X, EMD-1641). A comparison of the segmented maps corresponding to a single subunit is shown on top. A diameter comparison from the top view of the filaments is shown on the *bottom*. **b** Superimposition of the single L-type subunits: three from *B. subtilis* (*yellow*, S285P, S17P, E115G), one from *P. aeruginosa* (*orange*, G420A) and one from *S. enterica* (*dark green*, 3A5X). The *dashed line* indicates where the crystal structure of *S. enterica* ends. **c** Alignments of the flagellin amino acid sequence from four gram-positive bacteria and four gram-negative bacteria. Single mutants in *B. subtilis* are marked with filled *red stars*, the double mutants are marked with empty *red stars*, and the mutants in *P. aeruginosa* are marked with filled *blue squares*

helical parameters (rise and rotation), inter-subunit interactions and overall filament packing. This provides a comprehensive structural confirmation of the bi-state mechanism proposal[43], by using nine different structures from two different species.

**Domains and interfaces responsible for polymorphic switching.** Extensive work has been done in *S. enterica* to identify the residues responsible for the L- and R-type polymorphic switching, including mutagenesis[44], computational simulations[21] and cryo-EM approaches. To date, only two lower resolution cryo-EM

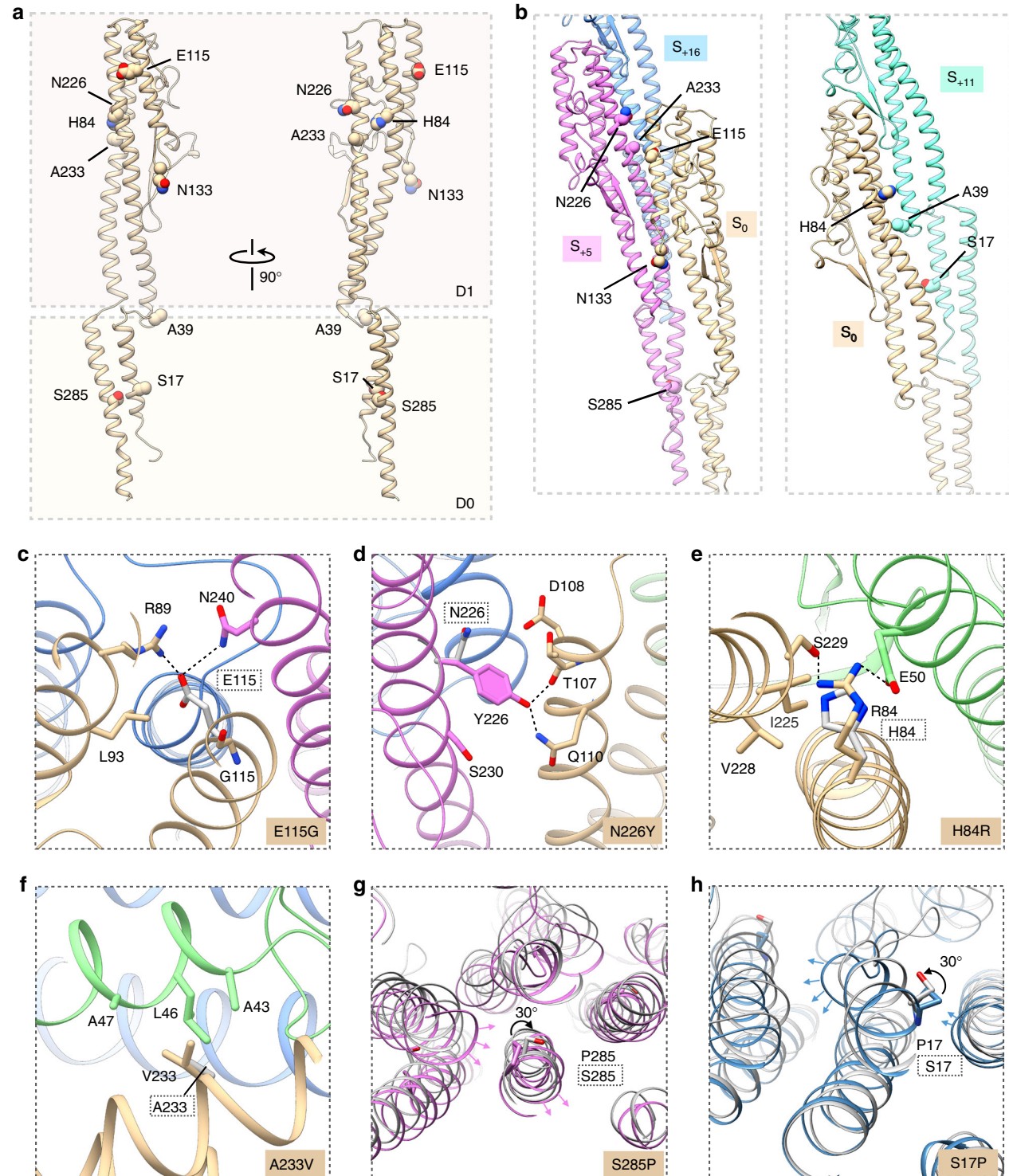

**Fig. 5** Mutation sites of *B. subtilis* and their molecular basis that lead to straight filaments. **a** Locating the mutation sites on the single flagellin subunit. **b** Locating the mutation sites on the 5-start interface (*left*) and the 11-start interface (*right*). **c–f** Subunits $S_0$, $S_{+5}$, $S_{+11}$ and $S_{+16}$ are colored in *dark yellow*, *purple*, *green* and *blue*, respectively. Un-mutated residues are shown in *gray*, and adapted from the other mutants in the same hand. Mutation sites for: left-handed mutation E115G **c**, right-handed mutation N226Y **d**, right-handed mutation H84R **e**, and right-handed mutation A233V **f**. **g** Top view of two mutant structures N226Y (*gray*) and S285P (*purple*) aligned by upper part of D0 domain (amino acids 18–32 and 268–284). **h** Top view of two mutant structures N226Y (*gray*) and S17P (*blue*) aligned by the same upper part of D0 domain

structures of *S. enterica* flagellar filaments have been reported[24, 25]. A comparison between *S. enterica*, *B. subtilis* and *P. aeruginosa* filament structures shows that their D0/D1 cores share a very conserved 11-protofilaments packing, with an inner diameter of ~ 25 Å and an outer diameter of ~ 125 Å (Fig. 4a). In fact, in *P. aeruginosa* we were able to generate the same-hand straight filaments using mutations corresponding to those reported in *S. enterica* (Fig. 2, Table 1). In contrast, in *Campylobacter jejuni* which has been reported to have seven and not 11 protofilaments, a mutation corresponding to the *Salmonella* A449V mutation does not lock the filaments into a straight form[27]. This strongly suggests that D0/D1 core is very conserved among species that have 11 protofilaments, and only the D0/D1 core is responsible for the L/R switching in these bacteria. To determine whether *S. enterica* shares the same D0/D1 core, five L-type subunits were superimposed in Fig. 4b. As mentioned previously, the three L-type subunits of *B. subtilis* and one L-type subunit of *P. aeruginosa* aligned very well in both D0 and D1 domains. However, the L-type subunit of *S. enterica* only aligns well with other subunits in most of its D1 domain (above the dashed line) where a crystal structure was available[26], while its D0 domain is shifted ~ 8 Å, presumably due to a poor map with very low resolution in this region (Supplementary Fig. 5) and the absence of a corresponding crystal structure. Thus, we present here the first accurate models of the D0-D1 connecting loop and the D0 domains.

In contrast to D0/D1, the D2/D3 domains are not conserved among species in terms of sequence, packing and oligomeric state within the filament (Figs. 4a, c). The flagellar filaments in *B. subtilis* and other Gram-positive species lack most of D2/D3, instead the β-hairpin and CD1 is connected by a small loop (Figs. 4a, c). On the other hand, *S. enterica*, *P. aeruginosa* and other gram-negative bacteria have D2/D3 domains, but with no detectable sequence identity (Supplementary Fig. 6). From our structures (Fig. 2d), in *P. aeruginosa* the D3 domain extends along the 11-start and forms a dimer with the D2 domain of another subunit; while in *S. enterica*, the D3 domain extends from the filament and does not interact with other subunits. Although the D2/D3 domains of Gram-negative bacteria contain ~ 200 amino acids, they may be considered unnecessary for the polymorphic switching for two reasons: (1) two spontaneous mutants of *S. typhimurium* previously reported with deletion of most or part of D2/D3 domains are still capable of swimming[45]; and (2) we show here in *B. subtilis* that switching occurs in the absence of D2/D3 domains (Figs. 4a, c).

During the random mutagenesis process to screen for straight mutants, we identified several new sites in *B. subtilis* not previously implicated in polymorphic switching (Fig. 4c). Strikingly, we found those sites are not necessarily conserved among different species or participate in unique inter-subunit interactions. These include three straight mutations outside of the D1 helices: S17P, S285P (both within D0) and the double mutant A39V/N133H (within the loop region) (Fig. 5a). We also identified several straight mutations that are only involved in the 11-start interactions: S17P, H84R and A39V in the double mutants (Fig. 5b). The existence of these mutations strongly suggests that the previous hypothesis, which proposes that only D1 domain interactions along the 5-start helices are involved in determining the L/R switching[14], is incomplete and over-simplified.

**Molecular basis for polymorphic switching**. From the analysis of seven straight mutants in *B. subtilis* and two straight mutants in *P. aeruginosa*, we confirmed the bi-state mechanism of polymorphic switching: that only two types of subunit-subunit

interactions (L-type and R- type) exist. This suggests that the wild type flagellar filaments with swimming motility must adopt an intricate "balance" in the flagellin sequence, so that the sequence does not generate a strong preference for either the L- or R- type conformation. Such a balance is required for the sharp transition of flagellar filaments switching between different waveforms during bacterial swimming and tumbling. The fact that straight phenotypes can be readily found due to single point mutations indicates that this balance is exquisitely sensitive to small changes, such as a single mutation, and can be easily tipped towards a dominant conformation, either all L or all R, which eliminates motility. To date, the molecular basis for the role of these mutations in abrogating switching has not been established.

To better understand the molecular basis of each mutation, we compared the local interactions of L- and R- type subunits in *B. subtilis* by examining ~ 20 amino acids around the mutation sites (Figs. 5c–f). In R-type subunits of *B. subtilis*, the side chain of E115 in $S_0$ forms two conserved hydrogen bonds with the side chain of N240 in subunit $S_{+5}$. These two sites (E115G and N240) are mostly conserved among different bacterial species (Fig. 4c), and we also detected the same interaction in the R-type A443V mutant in *P. aeruginosa*. In the *B. subtilis* E115G mutant, the filaments lose this conserved R-type interaction and therefore formed a left-handed filament (Fig. 5c). The residue N226 in *B. subtilis* is not a conserved site and does not form any conserved interactions with other subunits. In the R-type N226Y mutant, the side chain of Y226 in subunit $S_{+5}$ forms new hydrogen bonds with the side chain of Q110 and the main chain of T107 in subunit $S_0$. But in the L-type subunits, Y226 likely causes major steric clashes. Therefore, N226Y filaments adopt a right-handed form (Figs. 4c, 5d). Similarly, residue H84 in *B. subtilis* is not conserved in terms of sequence and subunit-subunit interactions. In the R-type mutation H84R, the R84 side chain in subunit $S_0$ forms new hydrogen bonds with the S229 side chain in subunit $S_0$ and the E50 side chain in subunit $S_{+11}$, enhancing the R-type interaction. As a result, the H84R filaments adopt a right-handed form (Figs. 4c, 5e). Another example is the R-type mutant A233V in *B. subtilis*. A233 is conserved among Gram-positive species but not in Gram-negative species. In the A233V mutant, valine in subunit $S_0$ makes stronger contacts than an alanine residue with the hydrophobic pockets formed by A43, L46 and A47 in subunit $S_{+11}$, and A233V filaments adopt a right-handed form (Figs. 4c, 5e). We also detected two mutations, S17P and S285P, located in the middle of the small coiled-coil D0 domain (Fig. 5a). These two residues are not conserved in other species, and proline is not found at this position in any wildtype species. This is likely because the proline cannot donate an amide hydrogen bond and its sidechain interferes sterically with the α-helical backbone. Therefore, mutation S17P or S285P in *B. subtilis* forces a local bend of ~ 30° in the helix axis, and this conformational change makes the D0 coiled-coil adopt an L-type form more easily (Figs. 5g, h).

**Designing mutations that lead to R-type transformation**. A stringent test of the model presented above is whether we can design new mutants that switch the wildtype filament into other waveforms. We hypothesized that constructing A237V or S71L mutants individually, or generating an A47CA233C double mutant (where an intermolecular disulfide would form between these residues) would shift filaments towards the R-type, while R91S or F132V individual mutants would shift filaments towards the L-type. These mutants were built and visualized by fluorescence microscopy (Figs. 6a, b).

Predictions were based on the comparison of the models of these mutations on the L- and R-type scaffolds (Supplementary

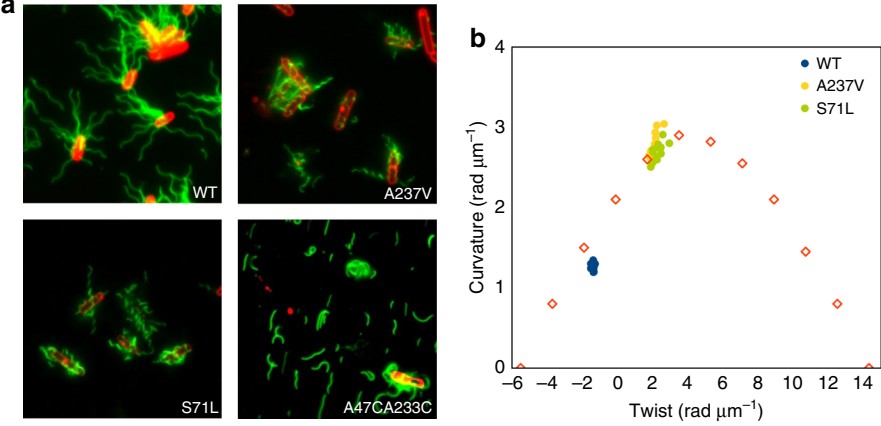

**Fig. 6** Predicted mutations of *B. subtilis* flagella filaments. **a** Fluorescence images of wildtype, curly form mutations A237V and S71L, and relatively straight form disulfide double mutant A47CA233C. **b** Plot of curvature κ against twist τ for measured helical forms of those *B. subtilis* mutants. The 12 discrete points corresponding to different theoretical waveforms in *S. enterica* are shown by *empty red squares*[14]

Fig. 7). For example, by introducing the A237V mutation into the L-type filament, it is likely that minor clashes will occur with E115 along the 5-start interface, while in the R-type filament this mutation will not clash with E115 and can form a stronger hydrophobic interaction with A118 (Figs. 6a, b, Supplementary Fig. 7a). Therefore, the A237V filament in theory prefers the R-state and this is consistent with the observed balance shifted to the right, from the wildtype to the curly form (Figs. 6a, b). Similarly, by introducing mutation S71L into the L-type filament, it likely introduces minor clashes with A259 along the 5-start interface, while in the R-type filament this mutation likely forms a better hydrophobic pocket with F132 and A259 (Figs. 6a, b, Supplementary Fig. 7b). To further explore our hypothesis, we designed a disulfide bond on the 11-start interface between residues A47 and A223. The $C_\alpha$ distance of those two residues in L-type and R-type filaments are ~ 7.3 Å and ~ 6.5 Å, respectively. By mutating both residues to cysteine, we predicted that the filaments would shift towards an R-state, as the $C_\alpha$ distance in the R-form is more favorable for forming a disulfide bond (Figs. 6a, b, Supplementary Fig. 7c). We observed that these filaments were very fragile and relatively straight (Fig. 6a). This is likely due to the state locked by the disulfide bond being slightly different from the preferred R-type filament conformation.

For the two L-type mutations we designed, we didn't observe any straight waveforms: one mutant (F132V) maintained the wildtype waveform and the other mutant (R91S) produced very short filaments (Supplementary Fig. 7d) so that the waveform cannot be determined. This is likely because there is only one intermediate waveform between wildtype and L-type straight form (while there are eight between wildtype and R-type straight form, Fig. 6b) and as a result it is harder to observe intermediate waveforms for L-type predictions. Combined with the R-type predictions, our results suggest that the waveform shift direction can be predicted based on the filament structures, but the degree of the shift cannot be predicted precisely.

## Discussion

Given the intensive study of bacterial filaments, we believe that we have now laid the foundation for many future studies of bacterial motility and polymorphic switching. The previous models for the L-[24] and R-state[25] of the *Salmonella* flagellar filament are useful in regions where crystal structures existed, but those models are much less accurate (Supplementary Fig. 5) in the D0 domain and the D0/D1 connecting loop where crystal

structures do not exist and the cryo-EM maps were limited in resolution. These incomplete structural models therefore led to inconsistent predictions at the single amino acid level[21]. Here our high-resolution reconstructions expand our knowledge and we can now observe how mutations in D0 can cause a switching of the flagellar structure.

Our observations do not support some predictions from previous studies. A molecular dynamics simulation of switching in *Salmonella* flagellar filaments[21] described three classes of inter-subunit residue pairs: "permanent" (those interactions that would be conserved between the L- and R-states), "sliding" (those variable hydrophobic or hydrophilic interactions with new partners that allow inter-subunit shear without a large change in energy), and "switching" (those key interactions that when made or broken cause a shift of the equilibrium from the L- to R-state or *vice versa*). We map the three classes proposed for *Salmonella* onto the corresponding observed residue pairs in *B. subtilis* (Supplementary Fig. 8) for both the L- (orange circles) and R-state (blue circles). For "permanent" interactions the distances should be conserved between the L- and R-states, while for both "sliding" and "switching" interactions the distances should differ between L and R. What we observe, however, is that some of the "sliding" and "switching" interactions are more conserved than the proposed "permanent" ones, and that most of the "permanent" ones are not permanent at all.

We have directly visualized how the outer D2/D3 domains within *P. aeruginosa* dimerize while not perturbing the conserved interactions that take place within the D0 and D1 core. However, the functional importance of D2/D3 domains still remains unknown, and they are dispensable in *Salmonella* for swimming motility[45]. Instead, they have been considered to contribute as radial spokes for bacteria that swim in a high viscosity environment[36] or provide antigenic variability used to escape immune surveillance[37].

The bacterial flagellar filament is an exquisitely tuned system that represents evolutionary development over hundreds of millions of years. In contrast to the simplicity of the bacterial flagellum, the eukaryotic flagellum, which has no homology to the bacterial one, is based upon microtubules and dynein rather than a homolog of flagellin and is currently estimated to contain more than 400 different proteins[46]. The archaeal flagellum, which has no homology to either the bacterial or eukaryotic ones, has only recently been solved at near-atomic resolution[47, 48] showing how the core of these archaeal filaments is formed by a domain that is homologous to the N-terminal domain of bacterial Type IV pilin.

Convergent evolution has thus yielded three very different flagellar filaments that all allow cells to swim, although by entirely different mechanisms. We are now entering a new era where the structure of such filaments can be solved at a near-atomic level of resolution using cryo-EM. We expect that the present study and future ones will yield new insights into how flagella-based swimming motility has independently arisen at least three different times in evolution using very different components, and how these convergent adaptations use very different mechanisms to achieve a similar function.

## Methods

**Strain and growth conditions.** *B. subtilis* strains were grown in lysogeny broth (LB) (10 g tryptone, 5 g yeast extract, 5 g NaCl per L) broth or on LB plates fortified with 1.5% Bacto agar at 37 °C. When appropriate, antibiotics were included at the following concentrations: 100 μg ml⁻¹ spectinomycin, 5 μg ml⁻¹ chloramphenicol, 5 μg ml⁻¹ kanamycin. Isopropyl β-D-thiogalactopyranoside (IPTG, Sigma) was added to the medium at the indicated concentration when appropriate.

For swimming motility assays, swim agar plates containing 25 ml of LB supplemented with 0.3% Bacto agar were freshly prepared. Plates were inoculated with a single overnight colony, incubated in a humidity chamber at 37 °C, and scored for motility after incubation for 18 h.

**Hag mutagenesis.** To produce the initial straight filament mutants, genomic DNA from DS1919 (*Δhag amyE::P_hag-hag^T209C spec*) was PCR amplified using primers 953/1009 and the Expand Long Template low-fidelity polymerase system (Roche) to introduce random base pair mutations into the *amyE::P_hag-hag^T209C spec* construct. PCR amplicons were transformed into DK620 strain deleted for *hag* in the highly competent PY79 lab strain background. Transformants were gridded on swim agar plates and swim-deficient mutants were isolated. The mutated *amyE::P_hag-hag^T209C spec* locus was then transduced to the DS1677 *Δhag* 3610 ancestral strain background via SPP1-mediated generalized phage transduction[49]. Strains were assessed for swarming motility (which like swimming motility requires functional flagella) as described below, and straight filament mutants were verified by fluorescent microscopy. Genomic DNA was isolated, and the *amyE::P_hag-hag^T209C spec construct* was amplified with primers 953/1009. The products were sequenced using primers 1008/3459. To increase flagellar expression, the *P_flache* promoter was replaced with an IPTG-inducible *P_hyspank* promoter by transducing the *P_flache ΩP_hyspank-fla/che operon* construct in each filament mutant background by SPP1-mediated generalized transduction using DK14 as a donor[50]. All strains used in this study are listed in Supplementary Table 2. All primers are listed in Supplementary Table 3.

For site-directed mutageneses, *hag* was inserted into the *aprE* ectopic locus by Gibson assembly[51]. The *aprE* upstream and downstream homology regions were amplified from 3610 chromosomal DNA as a template with primers 4440/4894 and 4439/4893, respectively. The *hag* gene and its native promoter were amplified with primers 4895/4932, and the kanamycin resistance cassette was amplified from pDG780 with primers 3251/4897[52]. Primers were designed such that 4439/4895, 3251/4932, and 4440/4897 have overlapping homology. All four fragments were ligated by Gibson assembly and transformed into the competent derivative of *B. subtilis* 3610, DK1042[53], to produce DK4136. DK4136 chromosomal DNA was isolated, and *hag* point mutants were generated by amplifying the *aprE::P_hag-hag kan* fragment with primers corresponding to the desired mutant in which the first primer number was paired with the *aprE* primer 4893 and the second primer number was paired with the *aprE* primer 4894 in PCR: Hag^S31P, 5518/5519; Hag^S71L, 5520/5521; Hag^R91S, 5522/5523; Hag^F123V, 5524/5525; Hag^A237V, 5526/5527; Hag^A47C, 5527/5528. The Hag^A47C A233C double mutant was generated by PCR amplifying *aprE::P_hag-hag^A47C kan* genomic DNA with primers 5530/5531 to introduce the A233C substitution. The primers containing the point mutant were complementary in sequence, and the two fragments were ligated by Gibson assembly and transformed into DK1042. Mutants were transduced into DK2790 by SPP1 generalized phage transduction to generate the strains used for experimentation.

**SPP1 phage transduction.** Serial ten-fold dilutions of SPP1 phage stock was added to 0.2 ml of dense culture grown in TY broth (autoclaved LB supplemented with 10 mM MgSO₄ and 100 μM MnSO₄) and incubated statically at 37 °C for 15 min. 3 ml of molten TY soft agar (TY supplemented with 0.5% Bacto agar) was added to the lysate, the mixture was poured onto fresh TY plates, and plates were incubated overnight at 30 °C. Top agar from plates producing plaques at high density was harvested in 5 ml TY, scraped into a 15 ml conical, vortexed to liberate phage, and centrifuged at 5000 x*g* for 10 min. The supernatant was passed through a 0.45 μm syringe filter, treated with 200 μl chloroform, sealed with parafilm, and stored at 4 °C.

Recipient cells were grown in TY broth at 37 °C to stationary. 1 ml of cells was treated with 25 μl of SPP1 donor phage stock, 9 ml of TY broth was added to the mixture, and cells were rocked at room temperature for 30 min. Transduced cells were centrifuged at 5000 *g* for 10 min, the supernatant was discarded and the supernatant was resuspended in the retained volume. The cell suspension was plated on LB plates supplemented with 1.5% Bacto agar, the appropriate antibiotic, and 10 mM sodium citrate.

**Swarm expansion assay.** Cells were grown in LB broth at 37 °C to mid-log phase. Cells were pelleted and resuspended to 10 OD₆₀₀ in PBS buffer (137 mM NaCl, 2.7 mM KCl, 10 mM Na₂HPO₄, and 2 mM KH₂PO₄, pH 8) containing 0.5% India ink (Higgins). LB plates containing 0.7% Bacto agar were freshly prepared (25 ml per plate) and dried in a laminar flow hood for 20 min. Dried plates were centrally inoculated with 10 μl of cell suspension, dried another 10 min, and incubated in a humidity chamber at 37 °C. Strains that did not swarm past the inoculation point (demarcated by the India ink) were saved for further experimentation.

**Fluorescence microscopy and waveform calculation.** For fluorescence microscopy of flagella, 1 mL of cell culture at mid-log phase was harvested in 1.5 ml microcentrifuge tubes at 7000 RPM for 1 min. The supernatant was discarded, pelleted cells were gently resuspended in 50 μL 1x PBS containing 5 μg ml⁻¹ Alexa Fluor 488 C₅ maleimide (Molecular Probes) to stain filaments with the Hag^T209C allele, and incubated in the dark at room temperature for 5 min. Cells were washed with 1 ml PBS, and cells were resuspended in 30 μl of PBS containing 5 μg ml⁻¹ FM4-64 for 3 min to stain membranes. Cells were washed once more with 1 ml PBS, resuspended in 30 μl of PBS, and 4 μl of suspension placed on a microscope slide and immobilized with a poly-L-lysine-treated coverslip.

Fluorescence microscopy was performed with a Nikon 80i microscope with a Nikon Plan Apo 100X objective and an Excite 120 metal halide lamp. FM4-64 was visualized with a C-FL HYQ Texas Red Filter Cube (excitation filter 532–587 nm, barrier filter > 590 nm). Alexa Fluor 488 was visualized using a C-FL HYQ FITC Filter Cube (FITC, excitation filter 460–500 nm, barrier filter 515–550 nm). Images were captured with a Photometrics Coolsnap HQ2 camera in black and white, false colored, and superimposed using Metamorph image software.

To identify the waveforms of wildtype, A237V, S71L and F123V mutants in *B. subtilis* under fluorescence microscopy, we used ImageJ to measure the length of the pitch and the diameter of the supercoiled filaments. To minimize errors in the measurement of the pitch, we particularly picked ten long supercoiled filaments for each strain. The twist and curvature plot shown in Fig. 6b was calculated from the pitch and diameter, using the previously established equations[54, 55].

**Purification of flagellar filaments of *Bacillus subtilis*.** Straight filament strains were transduced with DK14 to produce flagella overexpression constructs. Flagellar filaments were isolated using a modified protocol from Aizawa et al.[56]. Strains cultured overnight in 250 ml flasks were back-diluted to 0.1 OD₆₀₀ into 500 ml LB containing the proper antibiotic. Cultures were shaken at 37 °C to approximately 1.0 OD₆₀₀, harvested by centrifugation (6000 x*g* for 45 min), and gently resuspended in 20 ml of resuspension buffer (0.1 M Tris-HCl [pH 8.0], 0.5% Triton X-100). Cells were lysed by adding 2 ml of lysozyme (10 mg ml⁻¹ in milliQ water) and incubating at 37 °C with occasional stirring for 1 h. After lysis, 100 μl of DNase (1 mg mL⁻¹ in milliQ water) and 200 μl of 1 M MgCl₂ were added and the mixture was incubated at 37 °C for an additional 40 min to reduce the viscosity of the solution. Cell debris and intact cells were pelleted (10,000 *g*, 10 min), and the resulting supernatant was further subjected to high-speed centrifugation in a SW40 Ti ultracentrifuge rotor (100,000 *g*, 90 min). Pellets were resuspended in 20 ml of saline citrate (0.01 M sodium citrate, 0.1 M NaCl, pH 7.3), and proteins were precipitated by adding saturated ammonium sulfate solution to a 20% final concentration and gentle stirring. Precipitated protein was pelleted (3000 x*g*, 30 min) and resuspended in 11 ml sodium citrate, and 4.7 g of CsCl was added. The protein solution was ultracentrifuged in a SW40 Ti rotor (60,000 *g*, 18 h), and filaments formed an opaque white band approximately halfway down the tube. Filaments were collected by a 18ga needle and syringe, dialyzed in 0.1 M Tris-HCl buffer, and stored at 4 °C prior to imaging.

**Sample preparation of flagellar filaments of *P. aeruginosa*.** The FliC sequence from *Pseudomonas aeruginosa* PAO1 was aligned with FliC from *Salmonella typhimurium* to identify residues corresponding to the mutations G426A and A449V that were shown to result in straight filaments[11]. The *P. aeruginosa* FliC mutations G420A and A443V were cloned into pUCP20 and transformed into the transposon knockout line PW2971 (*ΔfliC*) obtained from the Manoil Lab at the University of Washington using electroporation as described[57]. *P. aeruginosa* PAO1 strains were grown overnight in LB liquid culture, cells were spun down and resuspended in PBS. Flagella were sheared off the cells by passing the suspension through a 21-gauge needle approximately 25 times. After centrifugation, the supernatant containing flagella was concentrated using a centricon.

**Cryo-electron microscopy and image processing.** Flagellar filament samples (3.5–4.5 μl) were applied to lacey carbon grids and vitrified with the Vitrobot Mark IV (FEI). The grids were imaged in a Titan Krios operating at 300 keV using a Falcon II camera with 1.05 Å per pixel sampling, with the imaging controlled by the EPU software. Images were collected using a defocus range of 0.5–3.0 μm, with a total exposure of 2 s dose-fractionated into seven chunks. All the images were first

motion corrected by the MotionCorr v2.1[58] and then the CTFFIND3 program[59] was used for determining the actual defocus of the images. Images with poor CTF estimation as well as defocus > 3 µm were discarded. Total number of images selected for each mutation are listed in Table 1. Filaments of varying lengths were boxed using the e2helixboxer program within EMAN2[60]. The SPIDER software package[61] was used for most other operations. The CTF was corrected by multiplying the images from the first two chunks (containing a dose of ~ 20 electrons per Å$^2$) with the theoretical CTF, which is a Wiener filter in the limit of a very poor signal-to-noise ratio (SNR). Overlapping long boxes (512 pixels for *bacillus subtilis* mutant N226Y, 384 pixels for the other mutants) with a shift of 7 pixels (~ 1.5 times of the axial rise) were cut from the long filaments. A reference-based sorting procedure was used to bin the segments based on the axial rise and azimuthal rotation. The number of total segments and the segments used after sorting for each mutation are listed in Table 1. After sorting, the segments were processed by the IHRSR method[62] to produce the final reconstructions.

**Model building**. The first model *B. subtilis* H84R was built using the *de novo* model-building protocol of Rosetta[31]. First, the map corresponding to a single flagellar subunit was segmented from the experimental filament density in Chimera[63]. Then a fragment library that contains pieces of experimentally determined structures was generated from the Robetta server[64]. This allowed for approximately 70% of the backbone of one H84R subunit being successfully built in Rosetta. Then the full-length H84R subunit was rebuilt with the RosettaCM protocol[31]. A total of 1000 full-length models were generated based on the segmented map, and the top ~ 15 models were selected according to the Rosetta's energy function. Those selected models were then combined into one model by manual editing in Coot[65] using the criteria of the local fit to the density map and the geometry statistics of the model. This model was used as the starting point for a whole filament RosettaCM rebuilding restrained by the determined helical symmetry and the whole experimental filament map. A total of 500 filament models were generated, and the best 15 models were combined again into a single model in Coot. Further refinements and editing were carried out by Phenix real-space refinement[66] and Coot. The other mutant filament models of *B. subtilis* were also built using H84R as the starting model, followed by the same RosettaCM, Coot editing and Phenix real-space refine protocol. The starting homology models of *P. aeruginosa* were generated by the I-TASSER server[67] using the same handed *B. subtilis* models as the templates, followed by the same RosettaCM, Coot editing and Phenix real-space refine protocol. MolProbity[68] was used to evaluate the quality of the models, and the statistics of all models are listed in Table 1.

**Negative stain imaging of flagellar filaments**. To examine the filament before cryo imaging, all flagellar filament samples were negatively stained. The filament samples (3.5 µl) were deposited for 30 s on a glow-discharged 300 mesh carbon-coated cooper grid (Electron Microscopy Sciences). The grid was then washed with 3 drops of 2% uranyl acetate. Images were taken with a Gatan CCD camera on a T12 microscope (FEI) operating at 80 keV. The images of negatively stained wild-type *P. aeruginosa* filaments were recorded with a 4k × 4k Gatan CCD camera on an F20 microscope (FEI) operating at 200 keV, in order to generate a power spectrum for analysis.

**Data availability**. The reconstruction maps were deposited in the Electron Microscopy Data Bank with accession numbers of 8847, 8848, 8849, 8850, 8851, 8852, 8853, 8855, and 8856. The corresponding filament models were deposited in the Protein Data Bank with accession numbers of 5WJT, 5WJU, 5WJV, 5WJW, 5WJX, 5WJY, 5WJZ, 5WK5 and 5WK6. The additional data that support the findings of this study are available from the corresponding author upon request.

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

## Acknowledgements

This work was supported by NIH GM122510 (to E.H.E.) and GM093030 (to D.B.K.). The cryo-EM work was conducted at the Molecular Electron Microscopy Core facility at the University of Virginia, which is supported by the School of Medicine and built with NIH grant G20-RR31199. The Titan Krios and Falcon II direct electron detector within the Core were purchased with NIH SIG S10-RR025067 and S10-OD018149, respectively. We thank Dr. Zhangli Su for helpful edits of the manuscript.

## Author contributions

A.M.B. performed site directed mutagenesis, phenotyping, and prepared the *B. subtilis* filament samples; R.E.C. screened for randomly generated non-motile alleles of *B. subtilis hag* and identified straight filament mutants; S.P. prepared the *P. aeruginosa* filament samples; A.O. and F.W. collected cryo-EM data; F.W. and E.H.E. performed image processing; F.W. did the structural modeling; F.W., A.M.B. and E.H.E. prepared figures; F.W. and E.H.E. wrote the manuscript; E.H.E., E.J.S. and D.B.K. conceived the study.

## Additional information

**Competing interests:** The authors declare no competing financial interests.

