## [Peer Review File · Nature Communications]

Reviewers' Comments:

Reviewer #1 (Remarks to the Author):

In this manuscript, Wang and colleagues present near atomic resolution structures of nine different types of straight flagellar filament from two bacterial species, *Bacillus subtilis* and *Pseudomonas aeruginosa*. The findings presented are novel and important as they reveal many new features of subunit packing that give novel insights into how flagellar filaments undergo the polymorphic transitions associated with rotation switching. The work is highly relevant to those working in the field of bacterial motility and, more broadly, the findings will be of interest to the wider community of researchers investigating the assembly and function of biological filaments.

Of particular note this work provides the first accurate model of the assembled flagellin D0 domain and the loop connecting D0 to the D1 domain. This model differs significantly from the previously published model based on lower resolution structures of straight filaments from *Salmonella enterica*. By solving structures of several filaments, each assembled from a different mutant flagellin, the authors reveal the importance of the loops, NL and CL, connecting D0 and D1 domains in causing the D1 domain to twist clockwise in L-subunits relative to the position of D1 in R-subunits. Further novel findings include the identification of additional new interaction interfaces in the D1 domain and a dimer interface between adjacent D2/D3 domains in the *Pseudomonas* filament.

The work confirms a bi-state mechanism underlying filament polymorphic transitions and goes on to identify additional residues important for polymorphic switching, including residues in D0, the D0-D1 connecting loop and residues involved in 11-start inter-subunit interactions. These new findings indicate that structural rearrangements of subunits during polymorphic switching are more complex than previously proposed. The authors test their model using site-directed mutagenesis, and the findings are highly convincing. All conclusions are supported by the data presented and I have no major criticisms.

Undoubtedly, this work will influence thinking in the field. Not only does it give a more detailed view of the structural basis of filament polymorphic switching, the data presented could also inform future work on flagellin subunit export and assembly.

Minor points

Abstract, line 4 – Should this read “Seven mutant flagellar filaments...” ?

Page 4 line 3 The sentence starting “The structure of...” is perhaps missing a word?

Page 27, line 2 “...dried...”

Reviewer #2 (Remarks to the Author):

This is a data-rich manuscript that describes the near-atomic resolution structure of a series of flagellar filaments from a gram positive (*B. subtilis*) and gram negative (*P. aeruginosa*) bacteria. These are the highest-resolution structures of straightened flagella to date. Overall the structural data is of good quality – my concerns are based on the presentation because it was difficult to keep track of 9 different sub 6.7 angstrom resolution structures. The findings are novel and of general interest to a broad audience and should be reconsidered upon revision, addressing the below concerns.

High resolution analysis of wild-type flagella is not possible because they are biochemically not straight – their function is to spin, forming a screw shape to generate force and allow swimming, switching, and swimming in a new direction to respond to environmental cues. To get around this, the researchers first used random mutagenesis to screen by fluorescence microscopy for single point variants of the *B. subtilis* filaments that abrogated bacterial motility by locking the filament in a single conformation. They isolated and then characterized 7 different locked point variants to between 3.8 and 6.7 angstroms resolution. Which variants are displayed in Figure 1?

They then recapitulated two *S. enterica* locked point variants in the *P. aeruginosa* flagella and determined one more structures of each hand. These filaments have a discontinuity owing to an insertion domain that violates the symmetry of the inner domains. They were able to assess the positions of these domains by reconstructing with a huge asymmetric unit but were unable to refine the insertion domain to high resolution. They show only the R-type structure in Figure 2 but the analysis in Figure 4 is performed on the L-type structure. Based on the FSC curve in the supplemental data and the RMSD analysis in Figure 3, these structures are of similar quality and show similar differences between the hands as in *B. subtilis*. Clarification of this choice would be welcome because displaying one structure in one figure and the other in the other figure seems a bit arbitrary and made it difficult to follow/assess what was being shown where.

After building pseudo atomic models of each, they were able to perform RMSD analysis to compare the folds of the flagellum monomer between the point variants of the same hand and the opposite hand as well as the domain that connects the two. They showed that the isolated

domains were quite similar even between the hands and the most significant variations were in the linker domain. Their additional resolution over previous *S. enterica* structures illuminated some over interpretations of those lower resolution models, in particular in the linker that joins the two conserved domains.

Finally, they used their structures to predict point variants that lock structures in either the R or L hand. They were successful at designing R-type filaments, based on fluorescence analysis, but not L-type filaments.

A few more comments/questions that need to be addressed:

1. The methods would benefit from a closer read. A few errors made it in the submission that I am certain will be tended to upon a resubmission (i.e. a reference to Tables X (strains) and Y (primers) – which don't seem to exist?; some typographical errors, dereied/should be dried; humid chamber should be humidity chamber). More critically, though, it was unclear to this reader exactly how the random mutagenesis screen worked to isolate individual clones of *B. subtilis*. There does not seem to be a step in which individual bacteria transductants that were identified as non-swarming were isolated into individual colonies.
2. The description of the rational mutations that were designed in *B. subtilis* in an attempt at predicting either the L or R-type structures could also benefit from some clarification. In general, it was difficult to understand what was done and what the result was. Some of this difficulty came from a reference to Figure 7A (which was 6A in reality) and some came because the methods for measuring curvature and twist in the generation of Figure 6B were not described. Were more mutations made that were predicted to be in L-state but were ambiguous like the R-state? If so, they need to be reported to put the predictions that worked as expected in context.
3. This same fluorescence assay was used to assess the *B. subtilis* motility-defective variants' suitability for high-resolution cryogenic 3DEM analysis but no raw data is shown. It would be interesting to know where these mutants fall on the plot in Figure 6B because it would show how well this low resolution structural assay reports on what is happening at the molecular level.
4. How did the researchers decide to use only the first 2/7 frames? Did inclusion of more affect resolution?
5. Some more details about the model building would also be welcome: why were only 15/1000 models selected to carry forward? By what criterion were they "manually combined?"
6. Why is the G420A *P. aeruginosa* structure at comparably high resolution with only 13% the number of sorted segments as *B. subtilis* A39VN133H, for example?
7. Some of the FSC curves show waviness at moderate resolutions (H84R, A39VN133H) that might indicate heterogeneity and imply overfitting. Does further sorting of the particles reduce this variability?
8. Throughout the manuscript, it is unclear from which structures displayed densities come. In addition to the above-mentioned confusion about figures 1-4, this information is not provided for

Figure S3. This information should be provided.

Reviewers' comments:

Reviewer #1 (Remarks to the Author):

In this manuscript, Wang and colleagues present near atomic resolution structures of nine different types of straight flagellar filament from two bacterial species, *Bacillus subtilis* and *Pseudomonas aeruginosa*. The findings presented are novel and important as they reveal many new features of subunit packing that give novel insights into how flagellar filaments undergo the polymorphic transitions associated with rotation switching. The work is highly relevant to those working in the field of bacterial motility and, more broadly, the findings will be of interest to the wider community of researchers investigating the assembly and function of biological filaments.

Of particular note this work provides the first accurate model of the assembled flagellin D0 domain and the loop connecting D0 to the D1 domain. This model differs significantly from the previously published model based on lower resolution structures of straight filaments from *Salmonella enterica*. By solving structures of several filaments, each assembled from a different mutant flagellin, the authors reveal the importance of the loops, NL and CL, connecting D0 and D1 domains in causing the D1 domain to twist clockwise in L-subunits relative to the position of D1 in R-subunits. Further novel findings include the identification of additional new interaction interfaces in the D1 domain and a dimer interface between adjacent D2/D3 domains in the *Pseudomonas* filament.

The work confirms a bi-state mechanism underlying filament polymorphic transitions and goes on to identify additional residues important for polymorphic switching, including residues in D0, the D0-D1 connecting loop and residues involved in 11-start inter-subunit interactions. These new findings indicate that structural rearrangements of subunits during polymorphic switching are more complex than previously proposed. The authors test their model using site-directed mutagenesis, and the findings are highly convincing. All conclusions are supported by the data presented and I have no major criticisms.

Undoubtedly, this work will influence thinking in the field. Not only does it give a more detailed view of the structural basis of filament polymorphic switching, the data presented could also inform future work on flagellin subunit export and assembly.

Minor points

Abstract, line 4 – Should this read “Seven mutant flagellar filaments...” ?

Page 4 line 3 The sentence starting “The structure of...” is perhaps missing a word?

Page 27, line 2 “...dried...”

We thank Reviewer #1 for the very strong support. The minor points raised were fixed in the revised manuscript.

Reviewer #2 (Remarks to the Author):

This is a data-rich manuscript that describes the near-atomic resolution structure of a series of flagellar filaments from a gram positive (*B. subtilis*) and gram negative (*P. aeruginosa*) bacteria. These are the highest-resolution structures of straightened flagella to date. Overall the structural data is of good quality – my concerns are based on the presentation because it was difficult to keep track of 9 different sub 6.7 angstrom resolution structures. The findings are novel and of general interest to a broad audience and should be reconsidered upon revision, addressing the below concerns.

High resolution analysis of wild-type flagella is not possible because they are biochemically not straight – their function is to spin, forming a screw shape to generate force and allow swimming, switching, and swimming in a new direction to respond to environmental cues. To get around this, the researchers first used random mutagenesis to screen by fluorescence microscopy for single point variants of the *B. subtilis* filaments that abrogated bacterial motility by locking the filament in a single conformation. They isolated and then characterized 7 different locked point variants to between 3.8 and 6.7 angstroms resolution. Which variants are displayed in Figure 1?

The variant displayed in Figure 1D is the left-handed mutant S285P in B. subtilis; the one in Figure 1E is right-handed mutant N226Y in B. subtilis. We also added this information into the Figure 1, Figure 2 and Figure S3 legends.

They then recapitulated two *S. enterica* locked point variants in the *P. aeruginosa* flagella and determined one more structures of each hand. These filaments have a discontinuity owing to an insertion domain that violates the symmetry of the inner domains. They were able to assess the positions of these domains by reconstructing with a huge asymmetric unit but were unable to refine the insertion domain to high resolution. They show only the R-type structure in Figure 2 but the analysis in Figure 4 is performed on the L-type structure. Based on the FSC curve in the supplemental data and the RMSD analysis in Figure 3, these structures are of similar quality and show similar differences between the hands as in *B. subtilis*. Clarification of this choice would

be welcome because displaying one structure in one figure and the other in the other figure seems a bit arbitrary and made it difficult to follow/assess what was being shown where.

We agree with the reviewer that the structures we solved are of similar quality for the L- and R- type. We would like to answer the reviewer's question in two regards:

(1) In Figure 2D, we were only showing the R-type structure, because we can only do a "non-helical perturbation" reconstruction on the R-type datasets. As shown in Table 1, cutting out 384-pixel long overlapping boxes with a shift of 7 pixel generates an L-type dataset having 17,450 segments, and R-type dataset having 209,905 segments before sorting. But if we want to do a reconstruction with the huge asymmetric unit (~100 Å rise) resulting from the non-helical perturbation, we need to cut out 640-pixel long overlapping boxes with a shift ~ 150 Å. In this case we have 8,481 segments for the R-type dataset, but for L-type we would only have 666 segments and this number of segments is not enough for a reasonable reconstruction.

*(2) In Figure 4, we compared the L-type structures from different species simply because the L-type filament of *S. enterica* was more recently published and its cryo-EM map was deposited (a map for the single subunit, not the whole filament). The R-type map from *Salmonella* is not available online in any form.*

After building pseudo atomic models of each, they were able to perform RMSD analysis to compare the folds of the flagellum monomer between the point variants of the same hand and the opposite hand as well as the domain that connects the two. They showed that the isolated domains were quite similar even between the hands and the most significant variations were in the linker domain. Their additional resolution over previous *S. enterica* structures illuminated some over interpretations of those lower resolution models, in particular in the linker that joins the two conserved domains.

Finally, they used their structures to predict point variants that lock structures in either the R or L hand. They were successful at designing R-type filaments, based on fluorescence analysis, but not L-type filaments.

A few more comments/questions that need to be addressed:

1. The methods would benefit from a closer read. A few errors made it in the submission that I am certain will be tended to upon a resubmission (i.e. a reference to Tables X (strains) and Y (primers) – which don't seem to exist?; some typographical errors,

dereied/should be dried; humid chamber should be humidity chamber). More critically, though, it was unclear to this reader exactly how the random mutagenesis screen worked to isolate individual clones of *B. subtilis*. There does not seem to be a step in which individual bacteria transductants that were identified as non-swarming were isolated into individual colonies.

We agree with the reviewer. The typographical errors have been fixed. Table S2 and Table S3 were added into the supplemental doc. And the “Hag mutagenesis” section in METHODS has been re-written.

2. The description of the rational mutations that were designed in *B. subtilis* in an attempt at predicting either the L or R-type structures could also benefit from some clarification. In general, it was difficult to understand what was done and what the result was. Some of this difficulty came from a reference to Figure 7A (which was 6A in reality) and some came because the methods for measuring curvature and twist in the generation of Figure 6B were not described. Were more mutations made that were predicted to be in L-state but were ambiguous like the R-state? If so, they need to be reported to put the predictions that worked as expected in context.

No more mutations were made than those described, and we already mentioned two predicted L-state in the main text. The methods for measuring curvature and twist have been added into the Methods as the reviewer suggested.

3. This same florescence assay was used to assess the *B. subtilis* motility-defective variants' suitability for high-resolution cryogenic 3DEM analysis but no raw data is shown. It would be interesting to know where these mutants fall on the plot in Figure 6B because it would show how well this low resolution structural assay reports on what is happening at the molecular level.

The straight mutants that we used for high-resolution cryo-EM have no curvature (a macroscopic quantity measured by light microscopy) and therefore all lie on the x-axis in the plot of Fig. 6B. We show here such a fluorescence image of a straight mutant below:

The fluorescence assay was simply used to determine that these non-motile mutants produce straight filaments. By cryo-EM, we can determine the twist of these mutants, and where they would lie on the x-axis in Figure 6B. The mutants have a twist of ~ -5.7 rad/ μm for the L-form and ~ 14.1 rad/ μm for the R-form.

4. How did the researchers decide to use only the first 2/7 frames? Did inclusion of more affect resolution?

At the electron dose we use including more chunks (where each “chunk” contains multiple frames with the Falcon II) will lead to the degradation of some sidechain densities. In general, we use all chunks to get a better CTF-estimation and a clearer image for boxing. But in reconstructions we only use the first two chunks to minimize radiation damage. A paper from our lab in press at Structure, “Refined cryo-EM structure of the T4 tail tube: exploring the lowest dose limit” shows the dose-dependent resolution, quite consistent with Grant and Grigorieff (2015).

5. Some more details about the model building would also be welcome: why were only 15/1000 models selected to carry forward? By what criterion were they “manually combined?”

More details were added into the Methods as the reviewer suggested.

(1) 1000 models were generated by RosettaCM based on the cryo-EM map. But most of the models are “bad” models, as judged by several energy and geometry terms, like Lazaridis-Karplus solvation energy, proline ring closure energy, hydrogen bond energy terms, disulfide bond energy term, Ramachandran preferences, omega angle preferences, etc. So we only picked the top ranked models based on these scores and started further refinement from them. The choice of 15 was arbitrary, and our final results would not have changed using 10 or 20.

(2) When all 15 models were examined it was clear that no single model was better than others everywhere. We usually find model A is better in one area but model B is better in some other area. “Better” is judged mainly by two factors: the local model:map correlation and the geometry statistics of the atomic model.

6. Why is the G420A *P. aeruginosa* structure at comparably high resolution with only 13% the number of sorted segments as *B. subtilis* A39VN133H, for example?

In most cases of cryo-EM reconstruction of helical filaments the resolution is not limited by the number of segments. It is more likely limited by the structural variability within the segments that we are analyzing. By reducing the length of the segments we can reduce the variability, but at the same time we reduce the ability to align segments accurately. The relationship between the resolution and the number of segments is not linear, but tends to follow a log-linear relation. The dataset of A39VN133H (4.3 Å) is the largest dataset we have, but the N226Y dataset reaches a higher resolution (3.8 Å) with only 50% the number of segments.

7. Some of the FSC curves show waviness at moderate resolutions (H84R, A39VN133H) that might indicate heterogeneity and imply overfitting. Does further sorting of the particles reduce this variability?

We do not think that the deviations from monotonicity in these plots suggests overfitting, as we have shown that the map:map FSC for A39VN133H actually yields a better resolution (4.1 Å) than the model:map FSC (4.3 Å). If we were overfitting, we would see the opposite, and the model:map FSC would appear to be significantly better than the map:map FSC! It is possible that the waviness results from heterogeneity, but further sorting reduces the size of the datasets and therefore lowers the overall resolution. We are also comparing atomic models “in vacuo” with real filaments surrounded by solvent. These solvent effects will appear differently at different resolutions: at low resolution they are reducing the overall contrast, while at very high resolution they can be ignored.

8. Throughout the manuscript, it is unclear from which structures displayed densities come. In addition to the above-mentioned confusion about figures 1-4, this information is not provided for Figure S3. This information should be provided.

We agree with the reviewer, and that information is now provided in the figure legends.

Reviewers' Comments:

Reviewer #2 (Remarks to the Author):

The authors have responded to my suggestions and the manuscript should be accepted for publication.